# QSGD: Communication-Efficient SGD via Gradient Quantization and Encoding

**Dan Alistarh**
IST Austria & ETH Zurich
dan.alistarh@ist.ac.at

**Demjan Grubic**
ETH Zurich & Google
demjangrubic@gmail.com

**Jerry Z. Li**
MIT
jerryzli@mit.edu

**Ryota Tomioka**
Microsoft Research
ryoto@microsoft.com

**Milan Vojnovic**
London School of Economics
M.Vojnovic@lse.ac.uk

## Abstract

Parallel implementations of stochastic gradient descent (SGD) have received significant research attention, thanks to its excellent scalability properties. A fundamental barrier when parallelizing SGD is the high bandwidth cost of communicating gradient updates between nodes; consequently, several lossy compresion heuristics have been proposed, by which nodes only communicate *quantized* gradients. Although effective in practice, these heuristics do not always converge.

In this paper, we propose *Quantized SGD (QSGD)*, a family of compression schemes with convergence guarantees and good practical performance. QSGD allows the user to smoothly trade off *communication bandwidth* and *convergence time*: nodes can adjust the number of bits sent per iteration, at the cost of possibly higher variance. We show that this trade-off is inherent, in the sense that improving it past some threshold would violate information-theoretic lower bounds. QSGD guarantees convergence for convex and non-convex objectives, under asynchrony, and can be extended to stochastic variance-reduced techniques.

When applied to training deep neural networks for image classification and automated speech recognition, QSGD leads to significant reductions in end-to-end training time. For instance, on 16GPUs, we can train the ResNet-152 network to full accuracy on ImageNet $1.8\times$ faster than the full-precision variant.

## 1 Introduction

The surge of massive data has led to significant interest in *distributed* algorithms for scaling computations in the context of machine learning and optimization. In this context, much attention has been devoted to scaling large-scale *stochastic gradient descent* (SGD) algorithms [33], which can be briefly defined as follows. Let $f : \mathbb{R}^n \to \mathbb{R}$ be a function which we want to minimize. We have access to stochastic gradients $\widetilde{g}$ such that $\mathbb{E}[\widetilde{g}(\boldsymbol{x})] = \nabla f(\boldsymbol{x})$. A standard instance of SGD will converge towards the minimum by iterating the procedure

$$\boldsymbol{x}_{t+1} = \boldsymbol{x}_t - \eta_t \widetilde{g}(\boldsymbol{x}_t), \tag{1}$$

where $\boldsymbol{x}_t$ is the current candidate, and $\eta_t$ is a variable step-size parameter. Notably, this arises if we are given i.i.d. data points $X_1, \ldots, X_m$ generated from an unknown distribution $D$, and a loss function $\ell(X, \theta)$, which measures the loss of the model $\theta$ at data point $X$. We wish to find a model $\theta^*$ which minimizes $f(\theta) = \mathbb{E}_{X \sim D}[\ell(X, \theta)]$, the expected loss to the data. This framework captures many fundamental tasks, such as neural network training.

In this paper, we focus on *parallel* SGD methods, which have received considerable attention recently due to their high scalability [6, 8, 32, 13]. Specifically, we consider a setting where a large dataset is partitioned among $K$ processors, which collectively minimize a function $f$. Each processor maintains a local copy of the parameter vector $x_t$; in each iteration, it obtains a new stochastic gradient update (corresponding to its local data). Processors then broadcast their gradient updates to their peers, and aggregate the gradients to compute the new iterate $x_{t+1}$.

In most current implementations of parallel SGD, in each iteration, each processor must communicate its entire gradient update to all other processors. If the gradient vector is dense, each processor will need to send and receive $n$ floating-point numbers *per iteration* to/from each peer to communicate the gradients and maintain the parameter vector $x$. In practical applications, communicating the gradients in each iteration has been observed to be a significant performance bottleneck [35, 37, 8].

One popular way to reduce this cost has been to perform *lossy compression of the gradients* [11, 1, 3, 10, 41]. A simple implementation is to simply *reduce precision* of the representation, which has been shown to converge under convexity and sparsity assumptions [10]. A more drastic quantization technique is *1BitSGD* [35, 37], which reduces each component of the gradient to *just its sign* (one bit), scaled by the average over the coordinates of $\widetilde{g}$, accumulating errors locally. 1BitSGD was experimentally observed to preserve convergence [35], under certain conditions; thanks to the reduction in communication, it enabled state-of-the-art scaling of deep neural networks (DNNs) for acoustic modelling [37]. However, it is currently not known if 1BitSGD provides any guarantees, even under strong assumptions, and it is not clear if higher compression is achievable.

**Contributions.** Our focus is understanding the trade-offs between the *communication cost* of data-parallel SGD, and its *convergence guarantees*. We propose a family of algorithms allowing for lossy compression of gradients called *Quantized SGD (QSGD)*, by which processors can trade-off the number of bits communicated per iteration with the variance added to the process.

QSGD is built on two algorithmic ideas. The first is an intuitive *stochastic quantization* scheme: given the gradient vector at a processor, we *quantize* each component by randomized rounding to a discrete set of values, in a principled way which preserves the statistical properties of the original. The second step is an efficient lossless code for quantized gradients, which exploits their statistical properties to generate efficient encodings. Our analysis gives tight bounds on the precision-variance trade-off induced by QSGD.

At one extreme of this trade-off, we can guarantee that each processor transmits at most $\sqrt{n}(\log n + O(1))$ expected bits per iteration, while increasing variance by at most a $\sqrt{n}$ multiplicative factor. At the other extreme, we show that each processor can transmit $\leq 2.8n + 32$ bits per iteration in expectation, while increasing variance by a only a factor of $2$. In particular, in the latter regime, compared to full precision SGD, we use $\approx 2.8n$ bits of communication per iteration as opposed to $32n$ bits, and guarantee at most $2\times$ more iterations, leading to bandwidth savings of $\approx 5.7\times$.

QSGD is fairly general: it can also be shown to converge, under assumptions, to local minima for non-convex objectives, as well as under asynchronous iterations. One non-trivial extension we develop is a *stochastic variance-reduced* [23] variant of QSGD, called QSVRG, which has exponential convergence rate.

One key question is whether QSGD's compression-variance trade-off is *inherent*: for instance, does any algorithm guaranteeing at most constant variance blowup need to transmit $\Omega(n)$ bits per iteration? The answer is positive: improving asymptotically upon this trade-off would break the communication complexity lower bound of distributed mean estimation (see [44, Proposition 2] and [38]).

**Experiments.** The crucial question is whether, in practice, QSGD can reduce communication cost by enough to offset the overhead of any additional iterations to convergence. The answer is yes. We explore the practicality of QSGD on a variety of state-of-the-art datasets and machine learning models: we examine its performance in training networks for image classification tasks (AlexNet, Inception, ResNet, and VGG) on the ImageNet [12] and CIFAR-10 [25] datasets, as well as on LSTMs [19] for speech recognition. We implement QSGD in Microsoft CNTK [3].

Experiments show that all these models can significantly benefit from reduced communication when doing multi-GPU training, *with virtually no accuracy loss*, and *under standard parameters*. For example, when training AlexNet on 16 GPUs with standard parameters, the reduction in communication time is $4\times$, and the reduction in training to the network's top accuracy is $2.5\times$. When training an LSTM on two GPUs, the reduction in communication time is $6.8\times$, while the reduction in training

time to the same target accuracy is $2.7\times$. Further, even computationally-heavy architectures such as Inception and ResNet can benefit from the reduction in communication: on 16GPUs, QSGD reduces the end-to-end convergence time of ResNet152 by approximately $2\times$. Networks trained with QSGD can converge to virtually the same accuracy as full-precision variants, and that gradient quantization may even slightly *improve* accuracy in some settings.

**Related Work.** One line of related research studies the *communication complexity* of convex optimization. In particular, [40] studied two-processor convex minimization in the same model, provided a lower bound of $\Omega(n(\log n + \log(1/\epsilon)))$ bits on the communication cost of $n$-dimensional convex problems, and proposed a *non-stochastic* algorithm for strongly convex problems, whose communication cost is within a log factor of the lower bound. By contrast, our focus is on *stochastic* gradient methods. Recent work [5] focused on *round complexity* lower bounds on the number of *communication rounds* necessary for convex learning.

Buckwild! [10] was the first to consider the convergence guarantees of low-precision SGD. It gave upper bounds on the error probability of SGD, assuming unbiased stochastic quantization, convexity, and gradient sparsity, and showed significant speedup when solving convex problems on CPUs. QSGD refines these results by focusing on the trade-off between communication and convergence. We view quantization as an independent source of variance for SGD, which allows us to employ standard convergence results [7]. The main differences from Buckwild! are that 1) we focus on the variance-precision trade-off; 2) our results apply to the quantized non-convex case; 3) we validate the practicality of our scheme on neural network training on GPUs. Concurrent work proposes TernGrad [41], which starts from a similar stochastic quantization, but focuses on the case where individual gradient components can have only three possible values. They show that significant speedups can be achieved on TensorFlow [1], while maintaining accuracy within a few percentage points relative to full precision. The main differences to our work are: 1) our implementation guarantees convergence under standard assumptions; 2) we strive to provide a black-box compression technique, with no additional hyperparameters to tune; 3) experimentally, QSGD maintains the same accuracy within the same target number of epochs; for this, we allow gradients to have larger bit width; 4) our experiments focus on the single-machine multi-GPU case.

We note that QSGD can be applied to solve the distributed mean estimation problem [38, 24] with an optimal error-communication trade-off in some regimes. In contrast to the elegant random rotation solution presented in [38], QSGD employs quantization and Elias coding. Our use case is different from the federated learning application of [38, 24], and has the advantage of being more efficient to compute on a GPU.

There is an extremely rich area studying algorithms and systems for efficient distributed large-scale learning, e.g. [6, 11, 1, 3, 39, 32, 10, 21, 43]. Significant interest has recently been dedicated to *quantized* frameworks, both for inference, e.g., [1, 17] and training [45, 35, 20, 37, 16, 10, 42]. In this context, [35] proposed 1BitSGD, a heuristic for compressing gradients in SGD, inspired by delta-sigma modulation [34]. It is implemented in Microsoft CNTK, and has a cost of $n$ bits and two floats per iteration. Variants of it were shown to perform well on large-scale Amazon datasets by [37]. Compared to 1BitSGD, QSGD can achieve asymptotically higher compression, provably converges under standard assumptions, and shows superior practical performance in some cases.

## 2    Preliminaries

SGD has many variants, with different preconditions and guarantees. Our techniques are rather portable, and can usually be applied in a black-box fashion on top of SGD. For conciseness, we will focus on a basic SGD setup. The following assumptions are standard; see e.g. [7].

Let $\mathcal{X} \subseteq \mathbb{R}^n$ be a known convex set, and let $f : \mathcal{X} \to \mathbb{R}$ be differentiable, convex, smooth, and unknown. We assume repeated access to stochastic gradients of $f$, which on (possibly random) input $\boldsymbol{x}$, outputs a direction which is in expectation the correct direction to move in. Formally:

**Definition 2.1.** Fix $f : \mathcal{X} \to \mathbb{R}$. A *stochastic gradient* for $f$ is a random function $\widetilde{g}(\boldsymbol{x})$ so that $\mathbb{E}[\widetilde{g}(\boldsymbol{x})] = \nabla f(\boldsymbol{x})$. We say the stochastic gradient has second moment at most $B$ if $\mathbb{E}[\|\widetilde{g}\|_2^2] \leq B$ for all $x \in \mathcal{X}$. We say it has variance at most $\sigma^2$ if $\mathbb{E}[\|\widetilde{g}(\boldsymbol{x}) - \nabla f(\boldsymbol{x})\|_2^2] \leq \sigma^2$ for all $x \in \mathcal{X}$.

Observe that any stochastic gradient with second moment bound $B$ is automatically also a stochastic gradient with variance bound $\sigma^2 = B$, since $\mathbb{E}[\|\widetilde{g}(\boldsymbol{x}) - \nabla f(\boldsymbol{x})\|^2] \leq \mathbb{E}[\|\widetilde{g}(\boldsymbol{x})\|^2]$ as long as $\mathbb{E}[\widetilde{g}(\boldsymbol{x})] = \nabla f(\boldsymbol{x})$. Second, in convex optimization, one often assumes a second moment bound

**Data**: Local copy of the parameter vector $\boldsymbol{x}$
1 **for** *each iteration t* **do**
2     Let $\widetilde{g}_t^i$ be an independent stochastic gradient ;
3     $\boxed{M^i \leftarrow \mathsf{Encode}(\widetilde{g}^i(\boldsymbol{x}))}$ //encode gradients ;
4     broadcast $M^i$ to all peers;
5     **for** *each peer $\ell$* **do**
6         receive $M^\ell$ from peer $\ell$;
7         $\boxed{\widehat{g}^\ell \leftarrow \mathsf{Decode}(M^\ell)}$ //decode gradients ;
8     **end**
9     $\boldsymbol{x}_{t+1} \leftarrow \boldsymbol{x}_t - (\eta_t/K) \sum_{\ell=1}^{K} \widehat{g}^\ell$;
10 **end**

**Algorithm 1:** Parallel SGD Algorithm.

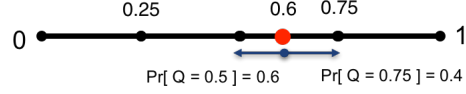

Figure 1: An illustration of generalized stochastic quantization with 5 levels.

when dealing with non-smooth convex optimization, and a variance bound when dealing with smooth convex optimization. However, for us it will be convenient to consistently assume a second moment bound. This does not seem to be a major distinction in theory or in practice [7].

Given access to stochastic gradients, and a starting point $\boldsymbol{x}_0$, SGD builds iterates $\boldsymbol{x}_t$ given by Equation (1), projected onto $\mathcal{X}$, where $(\eta_t)_{t \geq 0}$ is a sequence of step sizes. In this setting, one can show:

**Theorem 2.1** ([7], Theorem 6.3). *Let $\mathcal{X} \subseteq \mathbb{R}^n$ be convex, and let $f : \mathcal{X} \to \mathbb{R}$ be unknown, convex, and L-smooth. Let $\boldsymbol{x}_0 \in \mathcal{X}$ be given, and let $R^2 = \sup_{x \in \mathcal{X}} \|\boldsymbol{x} - \boldsymbol{x}_0\|^2$. Let $T > 0$ be fixed. Given repeated, independent access to stochastic gradients with variance bound $\sigma^2$ for $f$, SGD with initial point $\boldsymbol{x}_0$ and constant step sizes $\eta_t = \frac{1}{L+1/\gamma}$, where $\gamma = \frac{R}{\sigma}\sqrt{\frac{2}{T}}$, achieves*

$$\mathbb{E}\left[f\left(\frac{1}{T}\sum_{t=0}^{T}\boldsymbol{x}_t\right)\right] - \min_{\boldsymbol{x}\in\mathcal{X}}f(\boldsymbol{x}) \leq R\sqrt{\frac{2\sigma^2}{T}} + \frac{LR^2}{T} \ . \tag{2}$$

**Minibatched SGD.** A modification to the SGD scheme presented above often observed in practice is a technique known as *minibatching*. In minibatched SGD, updates are of the form $\boldsymbol{x}_{t+1} = \Pi_{\mathcal{X}}(\boldsymbol{x}_t - \eta_t \widetilde{G}_t(\boldsymbol{x}_t))$, where $\widetilde{G}_t(\boldsymbol{x}_t) = \frac{1}{m}\sum_{i=1}^{m}\widetilde{g}_{t,i}$, and where each $\widetilde{g}_{t,i}$ is an independent stochastic gradient for $f$ at $\boldsymbol{x}_t$. It is not hard to see that if $\widetilde{g}_{t,i}$ are stochastic gradients with variance bound $\sigma^2$, then the $\widetilde{G}_t$ is a stochastic gradient with variance bound $\sigma^2/m$. By inspection of Theorem 2.1, as long as the first term in (2) dominates, minibatched SGD requires $1/m$ fewer iterations to converge.

**Data-Parallel SGD.** We consider synchronous *data-parallel* SGD, modelling real-world multi-GPU systems, and focus on the communication cost of SGD in this setting. We have a set of $K$ processors $p_1, p_2, \ldots, p_K$ who proceed in synchronous steps, and communicate using point-to-point messages. Each processor maintains a local copy of a vector $\boldsymbol{x}$ of dimension $n$, representing the current estimate of the minimizer, and has access to private, independent stochastic gradients for $f$.

In each synchronous iteration, described in Algorithm 1, each processor aggregates the value of $\boldsymbol{x}$, then obtains random gradient updates for each component of $\boldsymbol{x}$, then communicates these updates to all peers, and finally aggregates the received updates and applies them locally. Importantly, we add *encoding* and *decoding* steps for the gradients before and after send/receive in lines 3 and 7, respectively. In the following, whenever describing a variant of SGD, we assume the above general pattern, and only specify the *encode/decode* functions. Notice that the decoding step does not necessarily recover the original gradient $\widetilde{g}^\ell$; instead, we usually apply an *approximate* version.

When the encoding and decoding steps are the identity (i.e., no encoding / decoding), we shall refer to this algorithm as *parallel SGD*. In this case, it is a simple calculation to see that at each processor, if $\boldsymbol{x}_t$ was the value of $\boldsymbol{x}$ that the processors held before iteration $t$, then the updated value of $\boldsymbol{x}$ by the end of this iteration is $\boldsymbol{x}_{t+1} = \boldsymbol{x}_t - (\eta_t/K)\sum_{\ell=1}^{K}\widetilde{g}^\ell(\boldsymbol{x}_t)$, where each $\widetilde{g}^\ell$ is a stochastic gradient. In particular, this update is merely a minibatched update of size $K$. Thus, by the discussion above, and by rephrasing Theorem 2.1, we have the following corollary:

**Corollary 2.2.** *Let $\mathcal{X}, f, L, \boldsymbol{x}_0$, and $R$ be as in Theorem 2.1. Fix $\epsilon > 0$. Suppose we run parallel SGD on $K$ processors, each with access to independent stochastic gradients with second moment*

*bound $B$, with step size $\eta_t = 1/(L + \sqrt{K}/\gamma)$, where $\gamma$ is as in Theorem 2.1. Then if*

$$T = O\left(R^2 \cdot \max\left(\frac{2B}{K\epsilon^2}, \frac{L}{\epsilon}\right)\right), \text{ then } \mathbb{E}\left[f\left(\frac{1}{T}\sum_{t=0}^{T}\boldsymbol{x}_t\right)\right] - \min_{\boldsymbol{x}\in\mathcal{X}} f(\boldsymbol{x}) \leq \epsilon. \qquad (3)$$

In most reasonable regimes, the first term of the max in (3) will dominate the number of iterations necessary. Specifically, the number of iterations will depend linearly on the second moment bound $B$.

# 3    Quantized Stochastic Gradient Descent (QSGD)

In this section, we present our main results on stochastically quantized SGD. Throughout, $\log$ denotes the base-2 logarithm, and the number of bits to represent a float is 32. For any vector $\boldsymbol{v} \in \mathbb{R}^n$, we let $\|\boldsymbol{v}\|_0$ denote the number of nonzeros of $\boldsymbol{v}$. For any string $\omega \in \{0,1\}^*$, we will let $|\omega|$ denote its length. For any scalar $x \in \mathbb{R}$, we let $\text{sgn}(x) \in \{-1, +1\}$ denote its sign, with $\text{sgn}(0) = 1$.

## 3.1    Generalized Stochastic Quantization and Coding

**Stochastic Quantization.** We now consider a general, parametrizable lossy-compression scheme for stochastic gradient vectors. The quantization function is denoted with $Q_s(\boldsymbol{v})$, where $s \geq 1$ is a tuning parameter, corresponding to the number of quantization levels we implement. Intuitively, we define $s$ uniformly distributed levels between 0 and 1, to which each value is quantized in a way which preserves the value in expectation, and introduces minimal variance. Please see Figure 1.

For any $\boldsymbol{v} \in \mathbb{R}^n$ with $\boldsymbol{v} \neq \boldsymbol{0}$, $Q_s(\boldsymbol{v})$ is defined as

$$Q_s(v_i) = \|\boldsymbol{v}\|_2 \cdot \text{sgn}(v_i) \cdot \xi_i(\boldsymbol{v}, s), \qquad (4)$$

where $\xi_i(\boldsymbol{v}, s)$'s are independent random variables defined as follows. Let $0 \leq \ell < s$ be an integer such that $|v_i|/\|\boldsymbol{v}\|_2 \in [\ell/s, (\ell+1)/s]$. That is, $[\ell/s, (\ell+1)/s]$ is the quantization interval corresponding to $|v_i|/\|\boldsymbol{v}\|_2$. Then

$$\xi_i(\boldsymbol{v}, s) = \begin{cases} \ell/s & \text{with probability } 1 - p\left(\frac{|v_i|}{\|\boldsymbol{v}\|_2}, s\right); \\ (\ell+1)/s & \text{otherwise.} \end{cases}$$

Here, $p(a, s) = as - \ell$ for any $a \in [0, 1]$. If $\boldsymbol{v} = \boldsymbol{0}$, then we define $Q(\boldsymbol{v}, s) = \boldsymbol{0}$.

The distribution of $\xi_i(\boldsymbol{v}, s)$ has minimal variance over distributions with support $\{0, 1/s, \ldots, 1\}$, and its expectation satisfies $\mathbb{E}[\xi_i(\boldsymbol{v}, s)] = |v_i|/\|\boldsymbol{v}\|_2$. Formally, we can show:

**Lemma 3.1.** *For any vector $\boldsymbol{v} \in \mathbb{R}^n$, we have that (i) $\mathbb{E}[Q_s(\boldsymbol{v})] = \boldsymbol{v}$ (unbiasedness), (ii) $\mathbb{E}[\|Q_s(\boldsymbol{v}) - \boldsymbol{v}\|_2^2] \leq \min(n/s^2, \sqrt{n}/s)\|\boldsymbol{v}\|_2^2$ (variance bound), and (iii) $\mathbb{E}[\|Q_s(\boldsymbol{v})\|_0] \leq s(s + \sqrt{n})$ (sparsity).*

**Efficient Coding of Gradients.** Observe that for any vector $\boldsymbol{v}$, the output of $Q_s(\boldsymbol{v})$ is naturally expressible by a tuple $(\|\boldsymbol{v}\|_2, \boldsymbol{\sigma}, \boldsymbol{\zeta})$, where $\boldsymbol{\sigma}$ is the vector of signs of the $v_i$'s and $\boldsymbol{\zeta}$ is the vector of integer values $s \cdot \xi_i(\boldsymbol{v}, s)$. The key idea behind the coding scheme is that not all integer values $s \cdot \xi_i(\boldsymbol{v}, s)$ can be equally likely: in particular, *larger integers are less frequent.* We will exploit this via a specialized *Elias* integer encoding [14], presented in full in the full version of our paper [4].

Intuitively, for any positive integer $k$, its code, denoted $\text{Elias}(k)$, starts from the binary representation of $k$, to which it prepends the *length* of this representation. It then recursively encodes this prefix. We show that for any positive integer $k$, the length of the resulting code has $|\text{Elias}(k)| = \log k + \log \log k + \ldots + 1 \leq (1 + o(1)) \log k + 1$, and that encoding and decoding can be done efficiently.

Given a gradient vector represented as the triple $(\|\boldsymbol{v}\|_2, \boldsymbol{\sigma}, \boldsymbol{\zeta})$, with $s$ quantization levels, our coding outputs a string $S$ defined as follows. First, it uses 32 bits to encode $\|\boldsymbol{v}\|_2$. It proceeds to encode using Elias recursive coding the position of the first nonzero entry of $\boldsymbol{\zeta}$. It then appends a bit denoting $\boldsymbol{\sigma}_i$ and follows that with $\text{Elias}(s \cdot \xi_i(\boldsymbol{v}, s))$. Iteratively, it proceeds to encode the distance from the current coordinate of $\boldsymbol{\zeta}$ to the next nonzero, and encodes the $\boldsymbol{\sigma}_i$ and $\boldsymbol{\zeta}_i$ for that coordinate in the same way. The decoding scheme is straightforward: we first read off 32 bits to construct $\|\boldsymbol{v}\|_2$, then iteratively use the decoding scheme for Elias recursive coding to read off the positions and values of the nonzeros of $\boldsymbol{\zeta}$ and $\boldsymbol{\sigma}$. The properties of the quantization and of the encoding imply the following.

**Theorem 3.2.** *Let $f : \mathbb{R}^n \to \mathbb{R}$ be fixed, and let $\boldsymbol{x} \in \mathbb{R}^n$ be arbitrary. Fix $s \geq 2$ quantization levels. If $\widetilde{g}(\boldsymbol{x})$ is a stochastic gradient for $f$ at $\boldsymbol{x}$ with second moment bound $B$, then $Q_s(\widetilde{g}(\boldsymbol{x}))$ is a*

stochastic gradient for $f$ at $\boldsymbol{x}$ with variance bound $\min\left(\frac{n}{s^2}, \frac{\sqrt{n}}{s}\right)B$. Moreover, there is an encoding scheme so that in expectation, the number of bits to communicate $Q_s(\widetilde{g}(\boldsymbol{x}))$ is upper bounded by

$$\left(3 + \left(\frac{3}{2} + o(1)\right)\log\left(\frac{2(s^2 + n)}{s(s + \sqrt{n})}\right)\right)s(s + \sqrt{n}) + 32.$$

**Sparse Regime.** For the case $s = 1$, i.e., quantization levels 0, 1, and $-1$, the gradient density is $O(\sqrt{n})$, while the second-moment blowup is $\leq \sqrt{n}$. Intuitively, this means that we will employ $O(\sqrt{n}\log n)$ bits per iteration, while the convergence time is increased by $O(\sqrt{n})$.

**Dense Regime.** The variance blowup is minimized to at most **2** for $s = \sqrt{n}$ quantization levels; in this case, we devise a more efficient encoding which yields an order of magnitude shorter codes compared to the full-precision variant. The proof of this statement is not entirely obvious, as it exploits both the statistical properties of the quantization and the guarantees of the Elias coding.

**Corollary 3.3.** *Let $f, \boldsymbol{x}$, and $\widetilde{g}(\boldsymbol{x})$ be as in Theorem 3.2. There is an encoding scheme for $Q_{\sqrt{n}}(\widetilde{g}(\boldsymbol{x}))$ which in expectation has length at most $2.8n + 32$.*

## 3.2 QSGD Guarantees

Putting the bounds on the communication and variance given above with the guarantees for SGD algorithms on smooth, convex functions yield the following results:

**Theorem 3.4** (Smooth Convex QSGD). *Let $\mathcal{X}, f, L, \boldsymbol{x}_0$, and $R$ be as in Theorem 2.1. Fix $\epsilon > 0$. Suppose we run parallel QSGD with $s$ quantization levels on $K$ processors accessing independent stochastic gradients with second moment bound $B$, with step size $\eta_t = 1/(L + \sqrt{K}/\gamma)$, where $\gamma$ is as in Theorem 2.1 with $\sigma = B'$, where $B' = \min\left(\frac{n}{s^2}, \frac{\sqrt{n}}{s}\right)B$. Then if $T = O\left(R^2 \cdot \max\left(\frac{2B'}{K\epsilon^2}, \frac{L}{\epsilon}\right)\right)$, then $\mathbb{E}\left[f\left(\frac{1}{T}\sum_{t=0}^{T}\boldsymbol{x}_t\right)\right] - \min_{\boldsymbol{x}\in\mathcal{X}} f(\boldsymbol{x}) \leq \epsilon$. Moreover, QSGD requires $\left(3 + \left(\frac{3}{2} + o(1)\right)\log\left(\frac{2(s^2+n)}{s^2+\sqrt{n}}\right)\right)(s^2 + \sqrt{n}) + 32$ bits of communication per round. In the special case when $s = \sqrt{n}$, this can be reduced to $2.8n + 32$.*

QSGD is quite portable, and can be applied to almost any stochastic gradient method. For illustration, we can use quantization along with [15] to get communication-efficient non-convex SGD.

**Theorem 3.5** (QSGD for smooth non-convex optimization). *Let $f : \mathbb{R}^n \to \mathbb{R}$ be a $L$-smooth (possibly nonconvex) function, and let $\boldsymbol{x}_1$ be an arbitrary initial point. Let $T > 0$ be fixed, and $s > 0$. Then there is a random stopping time $R$ supported on $\{1, \ldots, N\}$ so that QSGD with quantization level $s$, constant stepsizes $\eta = O(1/L)$ and access to stochastic gradients of $f$ with second moment bound $B$ satisfies $\frac{1}{L}\mathbb{E}\left[\|\nabla f(\boldsymbol{x})\|_2^2\right] \leq O\left(\frac{\sqrt{L(f(\boldsymbol{x}_1)-f^*)}}{N} + \frac{\min(n/s^2,\sqrt{n}/s)B}{L}\right)$. Moreover, the communication cost is the same as in Theorem 3.4.*

## 3.3 Quantized Variance-Reduced SGD

Assume we are given $K$ processors, and a parameter $m > 0$, where each processor $i$ has access to functions $\{f_{im/K}, \ldots, f_{(i+1)m/K-1}\}$. The goal is to approximately minimize $f = \frac{1}{m}\sum_{i=1}^{m} f_i$. For processor $i$, let $h_i = \frac{1}{m}\sum_{j=im/K}^{(i+1)m/K-1} f_j$ be the portion of $f$ that it knows, so that $f = \sum_{i=1}^{K} h_i$.

A natural question is whether we can apply stochastic quantization to reduce communication for parallel SVRG. Upon inspection, we notice that the resulting update will break standard SVRG. We resolve this technical issue, proving one can quantize SVRG updates using our techniques and still obtain the same convergence bounds.

**Algorithm Description.** Let $\widetilde{Q}(\boldsymbol{v}) = Q(\boldsymbol{v}, \sqrt{n})$, where $Q(\boldsymbol{v}, s)$ is defined as in Section 3.1. Given arbitrary starting point $\boldsymbol{x}_0$, we let $\boldsymbol{y}^{(1)} = \boldsymbol{x}_0$. At the beginning of epoch $p$, each processor broadcasts $\nabla h_i(\boldsymbol{y}^{(p)})$, that is, the unquantized full gradient, from which the processors each aggregate $\nabla f(\boldsymbol{y}^{(p)}) = \sum_{i=1}^{m} \nabla h_i(\boldsymbol{y}^{(p)})$. Within each epoch, for each iteration $t = 1, \ldots, T$, and for each processor $i = 1, \ldots, K$, we let $j_{i,t}^{(p)}$ be a uniformly random integer from $[m]$ completely independent from everything else. Then, in iteration $t$ in epoch $p$, processor $i$ broadcasts the update vector $\boldsymbol{u}_{t,i}^{(p)} = \widetilde{Q}\left(\nabla f_{j_{i,t}^{(p)}}(\boldsymbol{x}_t^{(p)}) - \nabla f_{j_{i,t}^{(p)}}(\boldsymbol{y}^{(p)}) + \nabla f(\boldsymbol{y}^{(p)})\right)$.

Table 1: Description of networks, final top-1 accuracy, as well as end-to-end training speedup on 8GPUs.

| Network | Dataset | Params. | Init. Rate | Top-1 (32bit) | Top-1 (QSGD) | Speedup (8 GPUs) |
|---|---|---|---|---|---|---|
| AlexNet | ImageNet | 62M | 0.07 | 59.50% | **60.05%** (4bit) | **2.05** $\times$ |
| ResNet152 | ImageNet | 60M | 1 | **77.0%** | 76.74% (8bit) | **1.56** $\times$ |
| ResNet50 | ImageNet | 25M | 1 | 74.68% | **74.76%** (4bit) | **1.26** $\times$ |
| ResNet110 | CIFAR-10 | 1M | 0.1 | 93.86% | **94.19%** (4bit) | **1.10** $\times$ |
| BN-Inception | ImageNet | 11M | 3.6 | - | - | **1.16** $\times$ (projected) |
| VGG19 | ImageNet | 143M | 0.1 | - | - | **2.25** $\times$ (projected) |
| LSTM | AN4 | 13M | 0.5 | 81.13% | **81.15** % (4bit) | **2** $\times$ (2 GPUs) |

Each processor then computes the total update $\boldsymbol{u}_t^{(p)} = \frac{1}{K}\sum_{i=1}^{K} \boldsymbol{u}_{t,i}$, and sets $\boldsymbol{x}_{t+1}^{(p)} = \boldsymbol{x}_t^{(p)} - \eta \boldsymbol{u}_t^{(p)}$. At the end of epoch $p$, each processor sets $\boldsymbol{y}^{(p+1)} = \frac{1}{T}\sum_{t=1}^{T}\boldsymbol{x}_t^{(p)}$. We can prove the following.

**Theorem 3.6.** *Let* $f(\boldsymbol{x}) = \frac{1}{m}\sum_{i=1}^{m} f_i(\boldsymbol{x})$, *where* $f$ *is* $\ell$-*strongly convex, and* $f_i$ *are convex and* $L$-*smooth, for all* $i$. *Let* $\boldsymbol{x}^*$ *be the unique minimizer of* $f$ *over* $\mathbb{R}^n$. *Then, if* $\eta = O(1/L)$ *and* $T = O(L/\ell)$, *then QSVRG with initial point* $\boldsymbol{y}^{(1)}$ *ensures* $\mathbb{E}\left[f(\boldsymbol{y}^{(p+1)})\right] - f(\boldsymbol{x}^*) \leq 0.9^p \left(f(\boldsymbol{y}^{(1)}) - f(\boldsymbol{x}^*)\right)$, *for any epoch* $p \geq 1$. *Moreover, QSVRG with* $T$ *iterations per epoch requires* $\leq (F + 2.8n)(T + 1) + Fn$ *bits of communication per epoch.*

**Discussion.** In particular, this allows us to largely decouple the dependence between $F$ and the condition number of $f$ in the communication. Let $\kappa = L/\ell$ denote the condition number of $f$. Observe that whenever $F \ll \kappa$, the second term is subsumed by the first and the per epoch communication is dominated by $(F + 2.8n)(T + 1)$. Specifically, for any fixed $\epsilon$, to attain accuracy $\epsilon$ we must take $F = O(\log 1/\epsilon)$. As long as $\log 1/\epsilon \geq \Omega(\kappa)$, which is true for instance in the case when $\kappa \geq \text{poly}\log(n)$ and $\epsilon \geq \text{poly}(1/n)$, then the communication per epoch is $O(\kappa(\log 1/\epsilon + n))$.

**Gradient Descent.** The full version of the paper [4] contains an application of QSGD to gradient descent. Roughly, in this case, QSGD can simply truncate the gradient to its top components, sorted by magnitude.

# 4   QSGD Variants

Our experiments will stretch the theory, as we use deep networks, with non-convex objectives. (We have also tested QSGD for convex objectives. Results closely follow the theory, and are therefore omitted.) Our implementations will depart from the previous algorithm description as follows.

First, we notice that the we can control the variance the quantization by quantizing into *buckets* of a fixed size $d$. If we view each gradient as a one-dimensional vector $\boldsymbol{v}$, reshaping tensors if necessary, a *bucket* will be defined as a set of $d$ consecutive vector values. (E.g. the $i$th bucket is the sub-vector $v[(i-1)d + 1 : i \cdot d]$.) We will quantize each bucket *independently*, using QSGD. Setting $d = 1$ corresponds to no quantization (vanilla SGD), and $d = n$ corresponds to full quantization, as described in the previous section. It is easy to see that, using bucketing, the guarantees from Lemma 3.1 will be expressed in terms of $d$, as opposed to the full dimension $n$. This provides a knob by which we can control variance, at the cost of storing an extra scaling factor on every $d$ bucket values. As an example, if we use a bucket size of $512$, and $4$ bits, the variance increase due to quantization will be upper bounded by only $\sqrt{512}/2^4 \simeq 1.41$. This provides a theoretical justification for the similar convergence rates we observe in practice.

The second difference from the theory is that we will scale by the *maximum* value of the vector (as opposed to the 2-norm). Intuitively, normalizing by the max preserves more values, and has slightly higher accuracy for the same number of iterations. Both methods have the same baseline bandwidth reduction because of lower bit width (e.g. 32 bits to 2 bits per dimension), but normalizing by the max no longer provides any sparsity guarantees. We note that this does not affect our bounds in the regime where we use $\Theta(\sqrt{n})$ quantization levels per component, as we employ no sparsity in that case. (However, we note that in practice max normalization also generates non-trivial sparsity.)

# 5   Experiments

**Setup.** We performed experiments on Amazon EC2 p2.16xlarge instances, with 16 NVIDIA K80 GPUs. Instances have GPUDirect peer-to-peer communication, but do not currently support NVIDIA

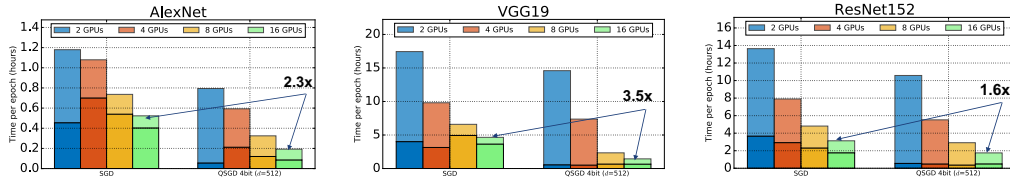

Figure 2: Breakdown of communication versus computation for various neural networks, on 2, 4, 8, 16 GPUs, for full 32-bit precision versus QSGD 4-bit. Each bar represents the total time for an epoch under standard parameters. Epoch time is broken down into *communication* (bottom, solid) and *computation* (top, transparent). Although epoch time *diminishes* as we parallelize, the proportion of communication *increases*.

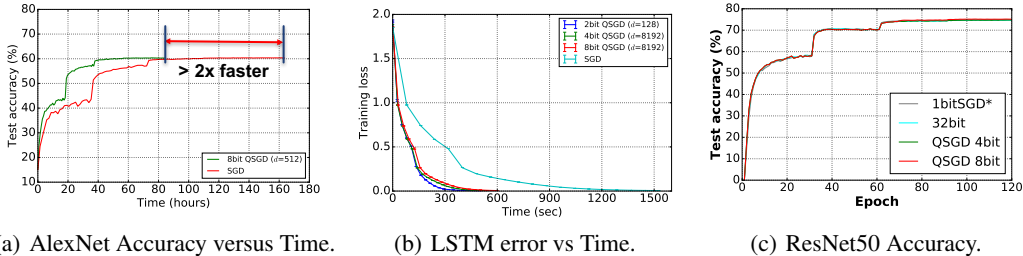

(a) AlexNet Accuracy versus Time.　　　(b) LSTM error vs Time.　　　(c) ResNet50 Accuracy.

Figure 3: Accuracy numbers for different networks. Light blue lines represent 32-bit accuracy.

**NCCL extensions.** We have implemented QSGD on GPUs using the Microsoft Cognitive Toolkit (CNTK) [3]. This package provides efficient (MPI-based) GPU-to-GPU communication, and implements an optimized version of 1bit-SGD [35]. Our code is released as open-source [31].

We execute two types of tasks: *image classification* on ILSVRC 2015 (ImageNet) [12], CIFAR-10 [25], and MNIST [27], and *speech recognition* on the CMU AN4 dataset [2]. For vision, we experimented with AlexNet [26], VGG [36], ResNet [18], and Inception with Batch Normalization [22] deep networks. For speech, we trained an LSTM network [19]. See Table 1 for details.

**Protocol.** Our methodology emphasizes *zero error tolerance*, in the sense that we always aim to preserve the accuracy of the networks trained. We used standard sizes for the networks, with hyper-parameters optimized for the 32bit precision variant. (Unless otherwise stated, we use the default networks and hyper-parameters optimized for full-precision CNTK 2.0.) We increased batch size when necessary to balance communication and computation for larger GPU counts, but never past the point where we lose accuracy. We employed *double buffering* [35] to perform communication and quantization concurrently with the computation. Quantization usually benefits from lowering learning rates; yet, we always run the 32bit learning rate, and decrease bucket size to reduce variance. We will not quantize small gradient matrices ($< 10K$ elements), since the computational cost of quantizing them significantly exceeds the reduction in communication. However, in all experiments, more than 99% of all parameters are transmitted in quantized form. We reshape matrices to fit bucket sizes, so that no receptive field is split across two buckets.

**Communication vs. Computation.** In the first set of experiments, we examine the ratio between computation and communication costs during training, for increased parallelism. The image classification networks are trained on ImageNet, while LSTM is trained on AN4. We examine the cost breakdown for these networks over a pass over the dataset (epoch). Figure 2 gives the results for various networks for image classification. The variance of epoch times is practically negligible (<1%), hence we omit confidence intervals.

Figure 2 leads to some interesting observations. First, based on the ratio of communication to computation, we can roughly split networks into *communication-intensive* (AlexNet, VGG, LSTM), and *computation-intensive* (Inception, ResNet). For both network types, the relative impact of communication *increases significantly* as we increase the number of GPUs. Examining the breakdown for the 32-bit version, all networks could significantly benefit from reduced communication. For

example, for AlexNet on 16 GPUs with batch size 1024, more than $80\%$ of training time is spent on communication, whereas for LSTM on 2 GPUs with batch size 256, the ratio is $71\%$. (These ratios can be slightly changed by increasing batch size, but this can decrease accuracy, see e.g. [21].)

Next, we examine the impact of QSGD on communication and overall training time. (Communication time includes time spent compressing and uncompressing gradients.) We measured QSGD with 2-bit quantization and 128 bucket size, and 4-bit and 8-bit quantization with 512 bucket size. The results for these two variants are similar, since the different bucket sizes mean that the 4bit version only sends $77\%$ more data than the 2-bit version (but $\sim 8\times$ less than 32-bit). These bucket sizes are chosen to ensure good convergence, but are not carefully tuned.

On 16GPU AlexNet with batch size 1024, 4-bit QSGD reduces communication time by $4\times$, and overall epoch time by $2.5\times$. On LSTM, it reduces communication time by $6.8\times$, and overall epoch time by $2.7\times$. Runtime improvements are non-trivial for all architectures we considered.

**Accuracy.** We now examine how QSGD influences accuracy and convergence rate. We ran AlexNet and ResNet to full convergence on ImageNet, LSTM on AN4, ResNet110 on CIFAR-10, as well as a two-layer perceptron on MNIST. Results are given in Figure 3, and exact numbers are given in Table 1. QSGD tests are performed on an 8GPU setup, and are compared against the best known full-precision accuracy of the networks. In general, we notice that 4bit or 8bit gradient quantization is sufficient to recover or even slightly improve full accuracy, while ensuring non-trivial speedup. Across all our experiments, 8-bit gradients with $512$ bucket size have been sufficient to recover or improve upon the full-precision accuracy. Our results are consistent with recent work [30] noting benefits of adding noise to gradients when training deep networks. Thus, quantization can be seen as a source of zero-mean noise, which happens to render communication more efficient. At the same time, we note that more aggressive quantization can hurt accuracy. In particular, 4-bit QSGD with $8192$ bucket size (not shown) loses $0.57\%$ for top-5 accuracy, and $0.68\%$ for top-1, versus full precision on AlexNet when trained for the same number of epochs. Also, QSGD with 2-bit and 64 bucket size has gap $1.73\%$ for top-1, and $1.18\%$ for top-1.

One issue we examined in more detail is which layers are more sensitive to quantization. It appears that quantizing *convolutional layers* too aggressively (e.g., 2-bit precision) can lead to accuracy loss if trained for the same period of time as the full precision variant. However, increasing precision to 4-bit or 8-bit recovers accuracy. This finding suggests that modern architectures for vision tasks, such as ResNet or Inception, which are almost entirely convolutional, may benefit less from quantization than recurrent deep networks such as LSTMs.

**Additional Experiments.** The full version of the paper contains additional experiments, including a full comparison with 1BitSGD. In brief, QSGD outperforms or matches the performance and final accuracy of 1BitSGD for the networks and parameter values we consider.

# 6 Conclusions and Future Work

We have presented QSGD, a family of SGD algorithms which allow a smooth trade off between the amount of communication per iteration and the running time. Experiments suggest that QSGD is highly competitive with the full-precision variant on a variety of tasks. There are a number of optimizations we did not explore. The most significant is leveraging the *sparsity* created by QSGD. Current implementations of MPI do not provide support for sparse types, but we plan to explore such support in future work. Further, we plan to examine the potential of QSGD in larger-scale applications, such as super-computing. On the theoretical side, it is interesting to consider applications of quantization beyond SGD.

The full version of this paper [4] contains complete proofs, as well as additional applications.

# 7 Acknowledgments

The authors would like to thank Martin Jaggi, Ce Zhang, Frank Seide and the CNTK team for their support during the development of this project, as well as the anonymous NIPS reviewers for their careful consideration and excellent suggestions. Dan Alistarh was supported by a Swiss National Fund Ambizione Fellowship. Jerry Li was supported by the NSF CAREER Award CCF-1453261, CCF-1565235, a Google Faculty Research Award, and an NSF Graduate Research Fellowship. This work was developed in part while Dan Alistarh, Jerri Li and Milan Vojnovic were with Microsoft Research Cambridge, UK.

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
