[Reviews · NeurIPS 2017]

Reviewer 1



Update: I decrease slightly the grade due to the mismatch between theoretical and practical results that could be better covered. Still this paper has strong experimental results and some theoretical results. I would encourage the authors to improve on the gap between the two. In this paper the author introduce Quantized SGD (QSGD), a scheme for reducing the communication cost of SGD when performing distributed optimization. The quantization scheme is useful as soon as one has to transmit gradients between different machines. The scheme is inspired by 1 BitSGD where 32 bits are used to transfer the norm of the gradient and then 1 bit per coordinate to encode the sign. The author extends this approach by allowing to send extra bits per coordinate to encode the scale of the coordinate. They obtain an unbiaised stochastic quantized version of the gradient. Therefore, one can still see QSGD as an SGD scheme, but with extra variance due to the quantization. Because of the quantization of the scale of each coordinate, small coordinates will be approximated by 0 and therefore the quantized gradient will be sparse, unlike 1BitSGD which gives a dense gradient. The authors use this property to further compress the gradient by encoding the non zero coordinates and only transfering information for those. In order to encode the position of those coordinates efficiently, the authors uses the Elias code, a variable length integer encoding scheme. The authors then applies this approach to deep learning model training. The authors also present how to quantize SVRG, a variance reduced SGD method with linear rate of convergence that can be used for convex optimization. Strengths ========= - The introduction is very clear and the authors do a great job at presenting the source of inspiration for this work, 1BitSGD. They also explain the limits of 1BitSGD (namely that there is no convergence guarentees) and clearly state the contribution of their paper (the ability to choose the number of bits and therefore the trade off between precision and communication cost, as well as having convergence guarantees). - the authors reintroduce the main results on SGD in order to properly understand the convergence of SGD in relation with the variance or second order moment of the gradients. This allow the reader to quickly understand how the quantization will impact convergence without requiring in depth technical knowledge of SGD. - As a consequence of the above points, this paper is very well self contained and can be read even with little knowledge about quantization or SGD training. - The authors did a massive investigation work, regarding both synchronous and asynchronous SGD, convex and non convex, as well as variance reduced SGD. Despite all this work, the authors did a good job at keeping the reading complexity low by separating properly each concern, sticking to simple case in the main paper (synchronous SGD) and keeping more complex details for the supplementary materials. - This paper contains a large amount of experiments, on different datasets, different architecture and comparing with existing methods such as 1bit SGD. - This paper offers a non trivial extension of 1bit SGD. For instance this method introduce sparsity in the gradient update while keeping convergence guarantees. Improvements ============ - The result on sparsity in Lemma 3.1, whose proof is in Lemma A.5 in the supplementary material seems wrong to me. Line 514 in the supplementary materials, first ui should be replaced by |ui| but more importantly, according to the definition on line 193 in the main paper, the probability is not |ui| but s|ui|. Thus, the sparsity bound becomes s2+ns. I would ask the authors to confirm this and update any conclusion that would be invalidated. This will have an impact on Lemma A.2. I would also encourage the authors to move this proof with the rest of the proof of Lemma 3.1. - Line 204, the description of Elias(k) is not super clear to me, maybe giving a few examples would make it better. The same applies to the description in the supplementary materials. Besides, the authors should cite the original Elias paper [1]. - The authors could comment on how far is the synchronous data-parallel SGD from current state of the art practice in distributed deep learning optimization. My own experience with mini-batches is that although theoretically and asymptotically using a batch size of B can make convergence take B times less iterations, this is not always true especially when far from the optimum, see [2] for instance, in practice the LR2T can dominate training for a significant amount of time. Other approaches than mini-batch exist, such as EASGD. I am not that familiar with the state of the art in distributed deep learning but I believe this paper could benefit from giving more details on the state of the art techniques and how QSGD can improve them (for instance for EASGD I am guessing that QSGD can allow for more frequent synchronization between the workers). Besides the model presented in this paper requires O(W^2) communication at every iteration where W is the number of worker, while only O(W) could be used with a central parameter server, thus this paper setup is particularly beneficial to QSGD. Such an approach is shortly described in the supplementary materials but as far as I could see there was no experiment with it. Overall this papers contains novel ideas, is serious and well written and therefore I believe it belongs at NIPS. It has detailed theoretical and experimental arguments in favor of QSGD. As noted above, I would have enjoyed more context on the impact on state of art distributed methods and there is a mistake in one of the proof that needs to be corrected. References ========== [1] Elias, P. (1975). Universal codeword sets and representations of the integers. IEEE transactions on information theory. [2] Sutskever, Ilya, et al. On the importance of initialization and momentum in deep learning. ICML 2013

Reviewer 2



Summary: This paper proposes a variant of SGD motivated by distributed implementation, where a novel compressing technique was introduced to quantize the stochastic gradient into a predefined range. It also analyzes the variance of the compressed SG and its affect on different stochastic algorithms. Strength: The proposed quantization technique is novel and leads to quantifiable trade-off between communication and convergence. Much experimental results on training CNN and RNN justify the proposed method. Questions/Comments: 1. In experiments, "When quantizing, we scale by the maximum value of the vector, ...". Is there any theoretical guarantee on using this? If yes, the paper should mention it before when presenting the encoding. 2. In experiments, "We do uno quantize very small gradient matrices in QSGD", how small is very small? 3. For QSVRG variant, it seems that the proof only holds for a smooth objective without a simple non-smooth regularizer. Can you extend the analysis for that case as well? I have read the authors' feedback. It is still not clear to me what guarantee that scaling by the max norm can provide.

Reviewer 3



Paper summary: This paper proposed a quantization based gradient compression approach for accelerating distributed/parallel stochastic gradient descent methods. The main idea is to quantize each component of the gradient vector by randomized rounding to a discrete set of values, followed by an Elias integer encoding scheme to represent and transit the quantized gradient. Theoretical analysis shows that the proposed method can considerably reduce the expected per-iteration bits of communication between machines/processors without sacrificing the convergence guarantee of exact SGD. The practicality of the proposed method is examined in training neural networks for image classification and speech recognition tasks. Comments: The paper is generally well organized and clearly presented. The empirical study is extensive and the reported results show some promise in achieving reasonable trade-off between efficiency and accuracy in a number of multi-GPU CNN training tasks. The reviewer has, however, several major concerns regarding its technical contents as in below: (1) The novelty of the proposed approach is limited. As acknowledged by the authors, the idea of gradient quantization and encoding has been extensively explored in neural networks acceleration and/or compression (see, e.g., [3, 13, 14, 17, 30, 35]). Although it is interesting to employ such a strategy on multi-GPU distributed learning tasks, the overall technical contribution in this line is rather marginal. Particularly, the quantization function Q_s(v) seems to be defined in a quite standard way; it is unclear why such a quantization scheme is more preferable than the state-of-the-arts. The used integer encoding method is also off-the-shelf. (2) The theoretical contribution is weak. The most interesting theoretical result of this paper is Theorem 3.2. The main message conveyed by this theorem is that by using the proposed quantization and encoding scheme, the per-iteration communication complexity is, roughly speaking, reduced by $\sqrt{n}$ while the variance of stochastic gradient is increased by $\sqrt{n}$. The reviewer is not convinced that such a result well justifies in theory the efficiency of method, as the improved communication efficiency is achieved at the price of dramatically increased iteration complexity. (3) There exists a gap between method description and implementation. The authors mentioned in line #283-284 that “When quantizing, we scale by the maximum value of the vector (as opposed to the 2-norm), as this simplifies computation and slightly reduces variance.”. It is a bit unclear to the reviewer why L2-norm (vector sum) is more computationally intensive than infinity-norm (vector sort) in the considered setting. Even if this is the case, is it possible to modify the theory part so as to justify such an implementation trick? (4) The paper title can be improved. The current title sounds too general and it should be modified to better reflect the “gradient quantization & encoding” nature of the proposed method.